# Characteristics of Retinitis Pigmentosa Associated with *ADGRV1* and Comparison with *USH2A* in Patients from a Multicentric Usher Syndrome Study Treatrush

**DOI:** 10.3390/ijms221910352

**Published:** 2021-09-26

**Authors:** Ana Fakin, Crystel Bonnet, Anne Kurtenbach, Saddek Mohand-Said, Ditta Zobor, Katarina Stingl, Francesco Testa, Francesca Simonelli, José-Alain Sahel, Isabelle Audo, Eberhart Zrenner, Marko Hawlina, Christine Petit

**Affiliations:** 1Eye Hospital, University Medical Centre Ljubljana, 1000 Ljubljana, Slovenia; marko.hawlina@gmail.com; 2Institut de l’audition, Institut Pasteur, Université de Paris, INSERM, 75012 Paris, France; crystel.bonnet@orange.fr (C.B.); christine.petit@pasteur.fr (C.P.); 3Institute for Ophthalmic Research, Centre for Ophthalmology, University of Tuebingen, 72076 Tuebingen, Germany; anne.kurtenbach@uni-tuebingen.de (A.K.); ditta.zobor@med.uni-tuebingen.de (D.Z.); ez@uni-tuebingen.de (E.Z.); 4Institut de la Vision, Sorbonne Université, INSERM, CNRS, 75012 Paris, France; saddekms@gmail.com (S.M.-S.); j.sahel@gmail.com (J.-A.S.); isabelle.audo@inserm.fr (I.A.); 5Centre Hospitalier National d’Ophtalmologie des Quinze-Vingts, Reference Center for Rare Disease REFERET, 75571 Paris, France; 6Center for Ophthalmology, University Eye Hospital, University of Tuebingen, 72076 Tuebingen, Germany; katarina.stingl@med.uni-tuebingen.de; 7Center for Rare Eye Diseases, University of Tuebingen, 72076 Tuebingen, Germany; 8Eye Clinic, Multidisciplinary Department of Medical, Surgical and Dental Sciences, University of Campania Luigi Vanvitelli, 80138 Naples, Italy; francesco.testa@unicampania.it (F.T.); francesca.simonelli@unicampania.it (F.S.); 9Department of Ophthalmology, Fondation Ophtalmologique Adolphe De Rothschild, 75019 Paris, France; 10Department of Ophthalmology, The University of Pittsburgh School of Medicine, Pittsburgh, PA 15213, USA; 11Collège de France, 75231 Paris, France

**Keywords:** Usher syndrome (USH), *USH2A*, *ADGRV1*, *VLGR1*, *GPR98*, *MASS1*, retinitis pigmentosa, fundus autofluorescence, hyperautofluorescent ring, adhesion G protein-coupled receptor V1

## Abstract

In contrast to *USH2A*, variants in *ADGRV1* are a minor cause of Usher syndrome type 2, and the associated phenotype is less known. The purpose of the study was to characterize the retinal phenotype of 18 *ADGRV1* patients (9 male, 9 female; median age 52 years) and compare it with that of 204 *USH2A* patients (111 male, 93 female; median age 43 years) in terms of nyctalopia onset, best corrected visual acuity (BCVA), fundus autofluorescence (FAF), and optical coherence tomography (OCT) features. There was no statistical difference in the median age at onset (30 and 18 years; Mann–Whitney U test, *p* = 0.13); the mean age when 50% of the patients reached legal blindness (≥1.0 log MAR) based on visual acuity (64 years for both groups; log-rank, *p* = 0.3); the risk of developing advanced retinal degeneration (patch or atrophy) with age (multiple logistic regression, *p* = 0.8); or the frequency of cystoid macular edema (31% vs. 26%, Fisher’s exact test, *p* = 0.4). *ADGRV1* and *USH2A* retinopathy were indistinguishable in all major functional and structural characteristics, suggesting that the loss of function of the corresponding proteins produces similar effects in the retina. The results are important for counseling *ADGRV1* patients, who represent the minor patient subgroup.

## 1. Introduction

Usher syndrome (USH) is a recessively inherited disorder characterized by the combination of retinal degeneration in the form of retinitis pigmentosa (RP), sensorineural hearing impairment, and, sometimes, vestibular dysfunction. Usher syndrome type 2 (USH2) is the most frequent subtype and presents with moderate to severe congenital hearing impairment, the onset of RP in the second decade of life, and normal vestibular function [1]. It has been associated with pathogenic variants in three genes: *USH2A*, *ADGRV1* (previously known as *VLGR1* or *GPR98*), and *WHRN*, identified in approximately 90%, 9%, and 1% of USH2 patients, respectively [2]. The pathogenic variants in *USH2A* are not only the most frequent cause of USH2 but also of RP without the associated hearing impairment (i.e., non-syndromic RP), accounting for 12–25% of cases [3,4]. The characteristics of retinal disease associated with *USH2A* have been described in many studies and are well known [5,6,7,8,9,10,11]. In contrast, the clinical presentations associated with *ADGRV1* and *WHRN* are less well known, as large patient cohorts have not yet been described phenotypically. The purpose of this study is to describe the characteristics of retinal disease in the largest cohort of *ADGRV1* patients to date and compare them with those of *USH2A* patients.

## 2. Results

### 2.1. Genetic Findings

The identified variants in *USH2A* and *ADGRV1* genes are listed in Appendix A. More than 1000 predicted pathogenic or likely pathogenic variants in the *USH2A* gene have been reported. Comparatively, only 168 predicted pathogenic or likely pathogenic variants in the *ADGRV1* gene have been listed in the LOVD3.0 database (https://databases.lovd.nl/shared/genes/GPR98; accessed on 15 August 2021). In this cohort, we identified a total of 27 different *ADGRV1* variants in 16 independent patients and 2 additional siblings. Twenty-four of these twenty-seven variants were loss of function variants (LOF) (i.e., twelve frameshift, nine nonsense, four splicing variants, and one large deletion), and three were missense variants. These variants were distributed along the *ADGRV1* gene. Of note, 78% of the patients carried biallelic LOF variants that are predicted to undergo nonsense-mediated decay (NMD) [12]. 

### 2.2. Disease Onset

There was no significant difference between the age at disease onset between *ADGRV1* (median 30 years, range 5–46 years) and *USH2A* patients (18 years, range 0–55 years) (Mann–Whitney U test, *p* = 0.13) (Figure 1). 

### 2.3. Visual Acuity

Table 1 shows the frequency of legal blindness in the better eye for each genotype. The multiple regression analysis showed a significant association between visual acuity in the better eye and age (*p* < 0.001) but not genotype (*p* = 0.3). The Kaplan–Meier survival analysis predicted that the mean age when 50% of the patients reached legal blindness based on visual acuity of 0.1 or less (1.0 logMAR or more) was 64 years for *USH2A* (95% CI 61–68 years) and 64 years for *ADGRV1* (95% CI 60–68 years) (Figure 2). The difference was not significant (log-rank, *p* = 0.3). The median value was 65 years for *USH2A* but could not be determined for *ADGRV1* due to the fact that all of the patients were censored prior (i.e., all had BCVA > 0.1 at the last follow-up). The median age when 50% of the patients were predicted to reach visual acuity <0.3 (>0.52) on the better eye was 57 and 54 years for *USH2A* and *ADGRV1*, respectively. The difference was not significant (log-ank, *p* = 0.9).

### 2.4. Fundus Autofluorescence Patterns

There was a good interocular symmetry of the FAF patterns; the same FAF pattern was present in 94% (136/145) of *USH2A* and 88% (14/16) of *ADGRV1* patients. The patients with asymmetry had either a combination of a hyperautofluorescent ring and patch (*n* = 7) or patch and atrophy (*n* = 4). Table 2 shows the distribution of the different FAF patterns between the two groups on the measured eye. The FAF patterns were qualitatively similar between the two groups (examples in Figure 3). A hyperautofluorescent ring was present in 70% of *USH2A* and 63% of *ADGRV1* patients. A multiple regression was performed to determine whether there was any difference in the autofluorescence patterns between different genetic groups with consideration of age-related disease progression. The multiple logistic regression showed an increased likelihood of advanced structural disease (patch or atrophy) with age (Exp(B) = 1.1, *p* < 0.001) but no correlation with gene (Exp(B) = 0.9, *p* = 0.8). Similarly, the multiple linear regression showed a significant association between ring diameter and age (*p* < 0.001) but not gene (*p* = 0.3). The survival analysis predicted that 50% of patients progress to advanced degeneration (patch or atrophy) at the median age of 58 years in both groups (95% CI 56–60 years for both) (Figure 4C).

### 2.5. Optical Coherence Tomography

CME was present in 26% (35/135) of *USH2A* and 31% (5/16) of *ADGRV1* patients with no statistical difference in frequency between the two groups (Fisher’s Exact Test, *p* = 0.4). Table 3 shows the number of patients with different degrees of CME. Considering all of the patients, CME was significantly more frequent in patients with a hyperautofluorescent ring pattern (34%, 30/87) than those with patch or atrophy (12%, 5/42) (Fisher’s Exact Test, *p* < 0.01). An analysis of foveal thickness (FT) with respect to age and genotype was performed on the largest group of patients imaged with the same OCT machine (Spectralis Heidelberg) to avoid measurement differences [13]. The patients with CME were excluded, leaving 83 *USH2A* and 10 *ADGRV1* patients. Multiple regression analysis showed a significant correlation between FT and age (*p* < 0.001), with FT decreasing with age, but no correlation with gene (*p* = 0.6) (Figure 5).

### 2.6. Correlation between Gender and Phenotype

The analysis of the same parameters was repeated within each genotypic group to determine whether there was any gender-based difference. There was no significant difference in disease onset, visual acuity, hyperautofluorescent ring diameter, or CME occurrence between male and female patients within each patient group.

### 2.7. Correlation between of Ethnicity and Phenotype

The study included patient cohorts from Germany, Italy, Spain, and Slovenia. Patient nationality was used as a proxy for ethnicity, and the Kruskal–Wallis test was used to determine whether it had any effect on the measured parameters. There was no significant difference in the median age at onset (17, 17, 19, and 17 years, respectively) and median ring size (4°, 4°, 3°, and 2°, respectively), while there was a significant difference in the median visual acuity (0.6, 0.2, 0.4, and 0.5 logMAR, respectively) and CME occurrence (27%, 10%, 43%, and 31%, respectively) (*p* < 0.05 for both parameters). There was a significant difference in the median age of the patients from different centers (44, 38, 42, and 53 years, respectively; *p* < 0.01).

## 3. Discussion

The present study includes the largest cohort of *ADGRV1* patients published to date (*n* = 18) and provides a detailed overview of the functional and structural parameters of the associated retinal disease. Importantly, data on visual acuity decline are useful in counseling patients. Furthermore, a comparison was made with a large *USH2A* cohort of patients with USH2, which revealed a striking similarity between the two phenotypes.

### 3.1. Genotype-Phenotype Correlations

*ADGRV1* variants are a rare cause of Usher syndrome, which is reflected by the scarcity of related phenotypic reports. The first paper on *ADGRV1* USH2 patients included 13 females from five families, with minimal phenotypic data [14]. The authors posed the question of possible sex predilection/bias towards females, but this was disputed by reports of patients of both sexes identified in four families [15,16,17], while one proposed that males may be more affected [16]. In the present study, the gender distribution was equal (nine male and nine female patients), and there was no significant difference in disease severity between the two genders. Therefore, the previous observations were likely due to chance and the small number of patients. There was, however, a considerable variability in the phenotype regardless of gender, which can be observed, for example, in the range of disease onset (Figure 1) and the diameter of the hyperautofluorescent ring (Figure 4B), suggesting that factors other than gender influence the disease expression.

The majority of the studies involving Usher syndrome patients with *ADGRV1* variants included ≤ 5 cases with no or minimal phenotypic data [14,15,16,17,18,19,20,21,22,23,24]. An exception is a report by Schwartz et al. [20], which investigated kinetic and chromatic perimetry, electrophysiology, and OCT parameters and provided a comparison with *USH2A*. However, the study was also hindered by the small number of *ADGRV1* patients (*n* = 3).

The present study thus provides the first comprehensive overview of *ADGRV1* retinopathy. The studied cohort of 18 patients exhibited retinitis pigmentosa, with the median onset at 30 years and a 50% chance of reaching legal blindness (VA ≤ 0.1) at 64 years of age. Similarly, the median age when 50% of the patients progressed to advanced macular degeneration (patch or atrophy autofluorescent patterns) was 58 years, which underscores the association between structure and function. 

A great similarity was observed in the comparison group consisting of 204 *USH2A* patients for all of the measured parameters. It is important to note, however, that only syndromic *USH2A* patients were included in this comparison as hearing loss was the inclusion criterion for this multicentric study. It has been shown that non-syndromic *USH2A* patients exhibit milder retinal disease, with delayed visual acuity loss and better-preserved amplitudes on electroretinography [6]. These differences in phenotype are probably related to the effects of different pathogenic variants, as null alleles have been associated with more severe hearing loss and retinal degeneration and are more often observed in syndromic cases [6,7,25,26]. However, a comparison of the structural measurements (ellipsoid zone line length) on retinal imaging between syndromic and non-syndromic USH2A patients showed no significant differences [8], and a large variability may be found even between patients with two null alleles [11]. 

The correlation between ethnicity and phenotype was examined in a limited fashion by using patient nationality as a proxy for ethnicity. There was no statistical difference in the median age at onset or median ring diameter, the two parameters that probably best describe the RP phenotype. There was a difference in CME occurrence; however, this could be affected by treatment (e.g., systemic acetazolamide, local acetazolamide, or observation). There was also a difference in the median visual acuity; however, this could be affected by opacities in the optic media (cataract or posterior capsular opacification), CME, or different methods of visual acuity measurement (Snellen or ETDRS). Therefore, the current study was not able to show a definitive correlation between ethnicity and phenotype.

### 3.2. Function of ADGRV1 and USH2A Proteins

Adhesion G protein-coupled receptor V1 (ADGRV1) and usherin (USH2A) are large proteins located at the periciliary region between the inner and outer segments of the photoreceptors and are thought to be important in stabilizing the connecting cilium. The proteins are anchored by a transmembrane domain and have long ectodomains that extend into the gap between the membranes of the connecting cilium and the apical inner segment, and they belong to the same protein complex [27,28,29,30,31,32]. 

The autosomal recessive inheritance of *USH2A* and *ADGRV1* suggests a loss of function mechanism. No gain of function variant has been reported in Usher syndrome genes, and no case of dominance has been observed, i.e., the parents of affected children are healthy. In support of this, most of the patients in this study harbored LOF variants. These included the frame-shifting variants as they were all predicted to undergo NMD (all were located >50 nucleotides from the last exon-exon junction), and thus expected to result in the absence of protein instead of an aberrant protein [12].

The indistinguishable clinical presentations of the two patient groups suggests that the loss of function of either of the proteins results in a similar functional defect in the retina. Considering the proteins’ shared location, there is a possibility that the proteins physically interact with each other and that the loss of function of either protein could result in a dysfunction of both. The possible interaction between USH2A and ADGRV1 has been schematically proposed by Maerker et al. [29] but has not yet been proven. It is also possible that the loss of either protein disrupts the interactions between other USH proteins in a similar manner. The existence of a quaternary USH2 complex has been proposed, where the third USH2 protein, whirlin (WHRN), and PDZD7, an USH2 modifier, are required for a complex formation with USH2A and ADGRV1. In this model, WHRN prefers to bind to USH2A, whereas PDZD7 prefers to bind to ADGRV1, while interaction between WHRN and PDZD7 is the bridge between USH2A and ADGRV1 [33].

Although both *USH2A* and *ADGRV1* are expressed in rods and cones, it is still not clear why their dysfunction primarily affects rods [29]. Considering that cones are shorter than rods, it is possible that the stabilization of the connecting cilium is not as important for their structural integrity.

### 3.3. Study Limitations

A limitation of the study is that only USH2A patients with hearing loss were included. Therefore, the findings relate only to those patients. This means that it is possible that there would be differences between *ADGRV1* and *USH2A* patients with non-syndromic RP. This does not exclude the differences between the RP of USH2 patients caused by *USH2A* or *ADGRVL1* defects and non-syndromic RP caused by *USH2A* defects. Another limitation is that only the retinal phenotype was studied. It would be interesting to expand the study to include hearing and vestibular impairment; however, this was beyond the scope of this study.

## 4. Materials and Methods

### 4.1. Patients

The study included 222 USH2 patients who were involved in multicentric study Treatrush [34]. The patients were from Germany (*n* = 72), France, (*n* = 69), Slovenia (*n* = 43), and Italy (*n* = 38). There were 204 *USH2A* patients (111 male; median age 43 years, range 1–80 years) and 18 *ADGRV1* patients (9 male; median age 52 years, range 23–68 years) with no statistical difference in the median age (Mann–Whitney U test, *p* = 0.11). Phenotypes of two patients (TR_EK-047 and TR_EK-048) have been described previously [16].

### 4.2. Genetic Analysis

Genetic analysis was performed by sequential use of three different techniques to analyze selected genomic regions: targeted exome sequencing, comparative genome hybridization, and quantitative exon amplification. The methods and detected variants were described in detail previously by Bonnet et al., Supplement 2 [34].

### 4.3. Clinical Assessment

Disease onset was defined as the age when patients or patients’ parents first reported night vision problems, and was determined from the hospital records (available for 148 patients). Visual function was measured with either Snellen (*n* = 116) or ETDRS (*n* = 83) and converted into logMAR equivalent. Counting finger and hand motion were quantified as 2.0 and 2.3 logMAR [35] and light perception as 2.8 logMAR [36]. Kaplan–Meier survival analysis was performed using best corrected visual acuity (BCVA) on the better eye, equal to or below 0.1 (≤6/60 or ≥1.0 logMAR; legal blindness) or below 0.3 (>6/20 or >0.52 logMAR; low vision), as a threshold. The better eye was used for the survival analysis as it was thought to best reflect the effect of the central visual loss on the patient’s daily life.

Fundus autofluorescence imaging (FAF) was performed in 161 patients using Spectralis imaging system (Heidelberg Engineering, Heidelberg, Germany). FAF pattern was classified as hyperautofluorescent ring, hyperautofluorescent patch, or atrophy, as described previously [9]. The inner horizontal diameter of the hyperautofluorescent ring was measured on FAF using ImageJ (imagej.nih.gov), where the 30° width of the squared FAF image was used as a reference (Figure 4D). The ring diameter was considered to be zero in cases of central patch or atrophy (examples in Figure 3 and Appendix A), as these patterns were previously shown to represent advanced retinal disease following the ring disappearance [9]. Optical coherence tomography (OCT) imaging was performed through the fovea in 151 patients using either Spectralis OCT (Heidelberg Engineering, Heidelberg, Germany; *n* = 122), Cirrus OCT (Carl Zeiss Meditec, Dublin, CA, USA; *n* = 23), or Stratus OCT (Carl Zeiss Meditec, Dublin, CA, USA; *n* = 6). The retinal thickness in the center of the fovea (FT) was measured on the OCT scan using ImageJ, where the 200 μm scale on the OCT image was used for reference. The presence or absence of cystoid macular edema (CME) was determined qualitatively. The right eye was used for analysis, except in seven cases, where poor image quality precluded the measurements.

### 4.4. Statistical Analysis

Statistical analysis was performed using SPSS software v. 22 (IBM SPSS Statistics; IBM Corporation, Chicago, IL, USA). The median values of the ages and ages at onset were compared using Mann–Whitney U test. Associations of different parameters with gene and age were tested using multiple linear regression (continuous variables) or multiple logistic regression (categorical variables). Fisher’s Exact test was used to compare the frequency of CME between the two patient groups. Kaplan–Meier survival curve was used to determine the mean and/or median age when 50% of patients reached low vision, legal blindness, and advanced retinal degeneration (FAF patterns of hyperautofluorescent patch or atrophy). Log-rank test was used to test for statistical differences between the survival curves of the two patient groups.

## 5. Conclusions

*ADGRV1* and *USH2A* retinopathy share all the major characteristics of functional and structural impairment, suggesting that the loss of function of the corresponding proteins produces similar effects in the retina. The results are important for counseling *ADGRV1* patients, who represent a minor patient subgroup.

## Figures and Tables

**Figure 1 ijms-22-10352-f001:**
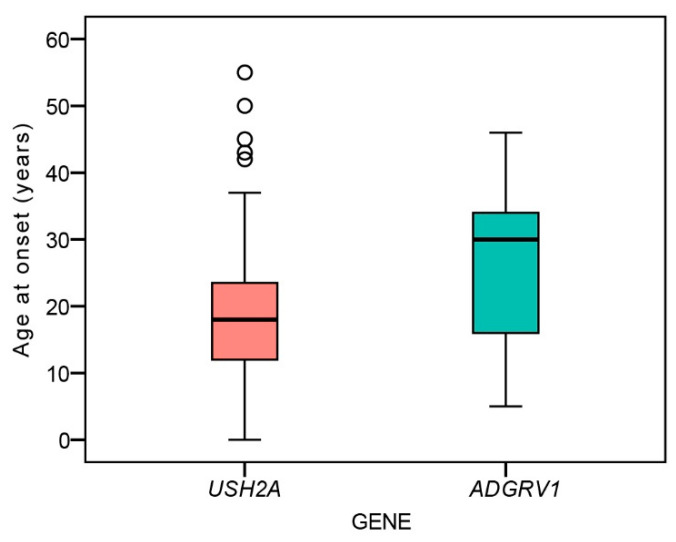
Boxplot chart showing the ages at onset of nyctalopia in the two patient groups. Horizontal lines represent the median values, boxes represent half of the data for each group/mutation, and whiskers represent the remaining data except in the case of the outliers (circles). Red color represents *USH2A* patients and green color represents *ADGRV1* patients.

**Figure 2 ijms-22-10352-f002:**
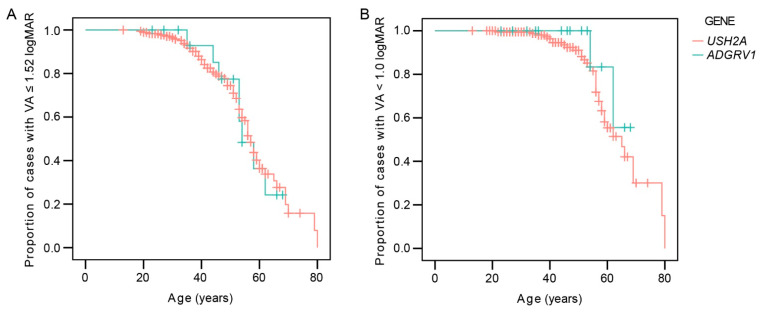
Kaplan–Meier survival analysis showing the ratio of patients reaching low vision (>1.52 logMAR or <6/20) (**A**) or legal blindness (≥1.0 log MAR or ≤6/60) (**B**). Red color represents *USH2A* patients and green color represents *ADGRV1* patients.

**Figure 3 ijms-22-10352-f003:**
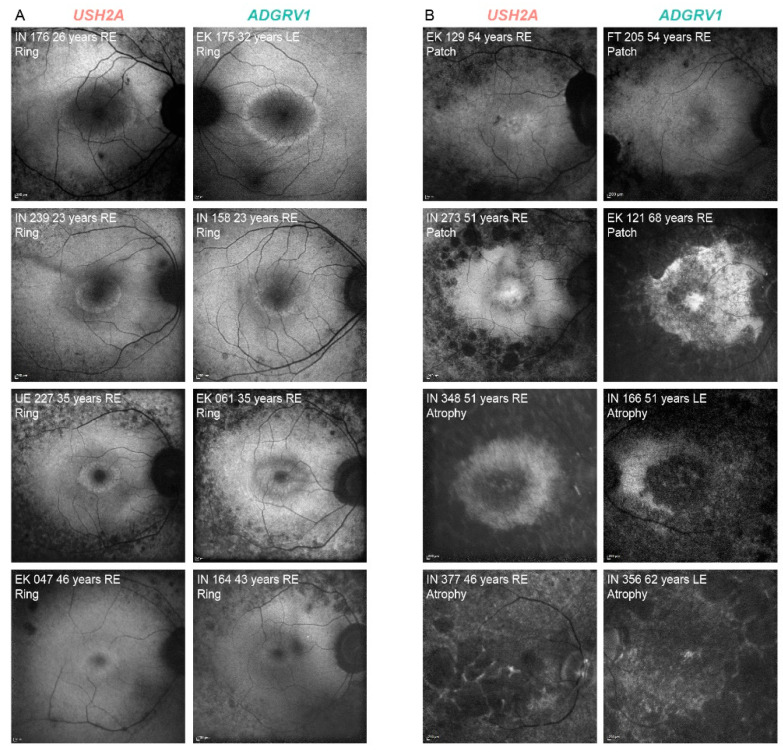
Representative FAF images of *USH2A* and *ADGRV1* patients. (**A**). hyperautofluorescent rings with preserved central retina. (**B**). advanced disease—hyperautofluorescent patch and atrophy. The patient’s ID, age and FAF pattern is stated in the top left corner of each image. The affected gene is stated above each column. Note that almost identical patterns were found in both groups. Red color represents *USH2A* patients and green color represents *ADGRV1* patients.

**Figure 4 ijms-22-10352-f004:**
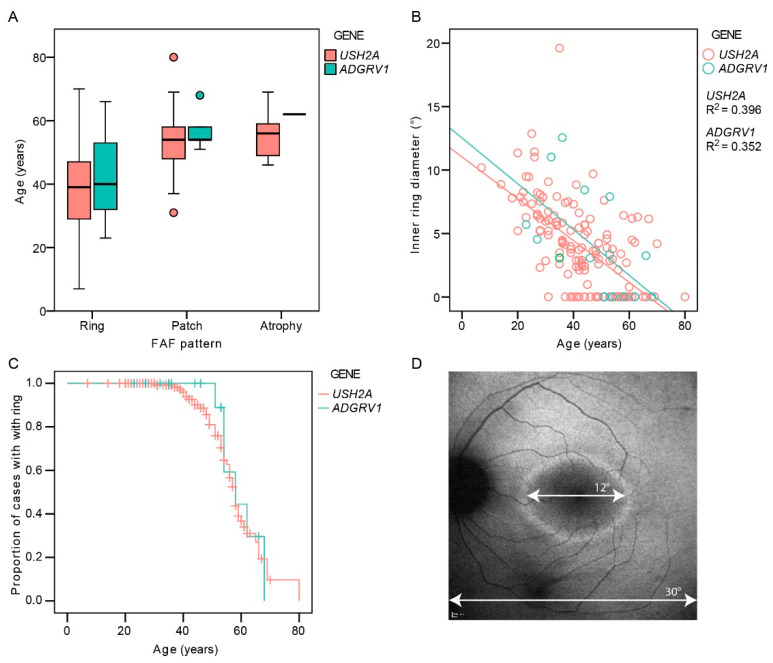
(**A**). Boxplot showing ages of patients with different autofluorescence patterns. (**B**). The diameter of the hyperautofluorescent ring in relation to age and genotype. (**C**). Kaplan–Meier survival analysis showing the proportion of cases with FAF pattern of hyperautofluorescent ring (preserved central retina) with increasing age. (**D**). Demonstration of the measurement of the inner ring diameter. Note the similarity between the survival curves representing the structural (**C**) and functional decline (Figure 2). Red color represents *USH2A* patients and green color represents *ADGRV1* patients.

**Figure 5 ijms-22-10352-f005:**
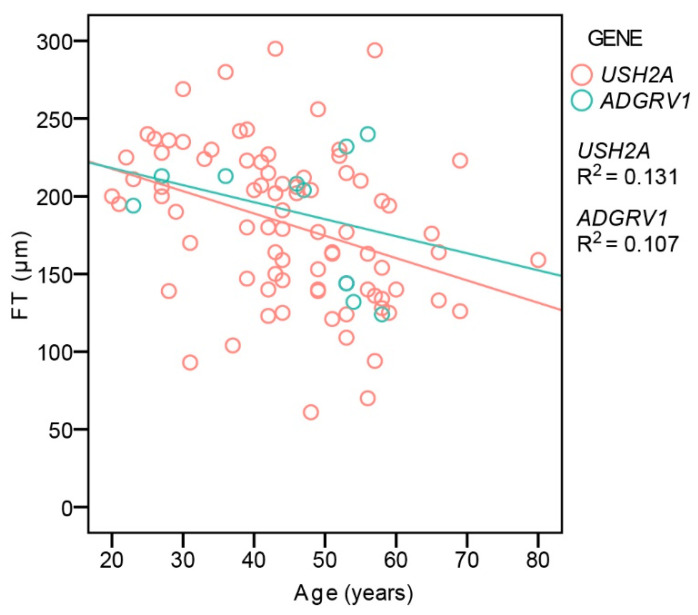
Central foveal thickness in relation to age and genotype. Red color represents *USH2A* patients and green color represents *ADGRV1* patients.

**Table 1 ijms-22-10352-t001:** Visual acuity.

	*USH2A*		*ADGRV1*	
BCVA	*n* (%)	Median Age (Range)	*n* (%)	Median Age (Range)
<1.0 logMAR (>6/60)	36 (20%)	31 years (13–70)	3 (18%)	27 years (23–36)
≥1.0 logMAR (≤6/60)	145 (80%)	47 years (18–80)	14 (82%)	53 years (32–68)
Total	181		17	

**Table 2 ijms-22-10352-t002:** Distribution of different FAF patterns in *USH2A* and *ADGRV1* patient groups.

	*USH2A*		*ADGRV1*	
FAF Pattern	*n* (%)	Median Age (Range)	*n* (%)	Median Age (Range)
Ring	102 (70%)	39 years (7–70)	10 (63%)	40 years (23–66)
Patch	33 (23%)	54 years (31–80)	5 (31%)	54 years (51–68)
Atrophy	10 (7%)	56 years (46–69)	1 (6%)	62 years
Total	145		16	

**Table 3 ijms-22-10352-t003:** Distribution of different degrees of CME in *USH2A* and *ADGRV1* patient groups.

	*USH2A*		*ADGRV1*	
CME	*n* (%)	Median Age (Range)	*n* (%)	Median Age (Range)
Absent	100 (74%)	44 years (20–80)	11 (69%)	51 years (23–58)
Minimal	26 (19%)	42 years (22–63)	3 (19%)	54 years (35–66)
Considerable	9 (7%)	50 years (26–70)	2 (13%)	38 years (32–44)
Total	135		16	

## Data Availability

The original data are available upon reasonable request to the corresponding author.

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
