# Peer review of "Characteristics of Retinitis Pigmentosa Associated with ADGRV1 and Comparison with USH2A in Patients from a Multicentric Usher Syndrome Study Treatrush"

_ijms, 2021, doi:10.3390/ijms221910352_

Round 1

Reviewer 1 Report

In this study, the authors found ADGRV1 and USH2A retinopathy were indistinguishable in all major functional and structural characteristics, suggesting that the loss of function of the corresponding proteins produces similar effects in the retina. The results are important for counseling ADGRV1 patients who represent the minor patient subgroup. These are important findings for counseling ADGRV1 patients who represent a minor patient subgroup.

Author Response

Thank you for the positive review!

Reviewer 2 Report

This manuscript by Fakin et al. investigated the characteristics of retinitis pigmentosa associated with ADGRV1 and comparison with USH2A in patients from a multicentric Usher syndrome study Treatrush. The authors used multiple techniques and appropriate statistical analyses.  It is suitable for publication in IJMS with minor revision. I only have some minor questions that need to be addressed before publication.

 .

  1. The authors showed that there were no significant differences between the age at disease onset as well as gender on phenotype between ADGRV1 and USH2A patients.

I was wondering whether there was any racial related difference in your study. I understand that the small number of cases carrying an ADGRV1 mutation may be not enough to draw any conclusion on this issue.

  1. The authors concluded that all the USH2A and ADGRV1 mutations in patients were the loss of function mutations. I was wondering whether there is any possibility that some frameshift mutations in these genes may ass some artificial amino acids in coding region and cause the mutant protein retaining in ER and act as a gain of function mutation.
  2. Is there any possibly that USH2A and ADGRV1proteins may physically interacted or USH2A or ADGRV1 may have some protein-protein interacting binding partners in retinal neurons, and the mutant proteins may cause these protein-protein interactions interrupted and cause the loss function of their interacting proteins. If so, please give a brief discussion.
  3. On line 92, the median value was 65 years for USH2A, but could be determined for ADGRV1 due to …, is it possible that “but could be determined …’ should be changed to “but could not be determined …”?

Thanks for the invitation!

Author Response

Thank you for the positive review and the points raised which improved the discussion. Please see the answers below.

Point 1: The authors showed that there were no significant differences between the age at disease onset as well as gender on phenotype between ADGRV1 and USH2A patients.

I was wondering whether there was any racial related difference in your study. I understand that the small number of cases carrying an ADGRV1 mutation may be not enough to draw any conclusion on this issue.

Response 1: Thank you for an interesting question. The number od ADGRV1 patients was indeed to small to perform calculations within the group, however as the phenotypes of USH2A and ADGRV1 were so similar, we performed this analysis on the whole cohort. The following paragraphs were added to the results and discussion.

Results:

2.7 Correlation between of ethnicity and phenotype

Study included patient cohorts from Germany, Italy, Spain and Slovenia. Patient nationality was used as a proxy for ethnicity and Kruskal-Wallis test was used to determine whether it had any effect on the measured parameters. There was no significant difference in the median age at onset (17, 17, 19, and 17 years, respectively) and median ring size (4°, 4°, 3° and 2°, respectively), while there was a significant difference in the median visual acuity (0.6, 0.2, 0.4, and 0.5 logMAR, respectively) and CME occurrence (27 %, 10 %, 43 % and 31 %, respectively) (p < 0.05 for both parameters). There was a significant difference in the median age of the patients from different centre (44, 38, 42, and 53 years, respectively; p < 0.01).

Discussion: There correlation between ethnicity on phenotype was examined in a limited fashion by usint patinent nationality as a proxy for ethnicity. There was no statistical difference in the median age at onset or median ring diameter, the two parameters that probably best desribe the RP phenotype. There was a difference in CME ocurance, however this could be affected by treatment (e.g. systemmic acetazolamide, local acetazolamide, or observation). There was also a difference in the median visual acuity, however this could be affected by opacities in the optic media (cataract or posterior capsular opacification), CME, or different methods of visual acuity measurement (Snellen or ETDRS). Therefore, the current study was not able to show a definitive correlation between ethnicity and phenotype.

Point 2: The authors concluded that all the USH2A and ADGRV1 mutations in patients were the loss of function mutations. I was wondering whether there is any possibility that some frameshift mutations in these genes may ass some artificial amino acids in coding region and cause the mutant protein retaining in ER and act as a gain of function mutation.

Answer 2: Thank you for the comment. All frame-shifting mutations were predicted to undergo nonsense mediated decay and thus result in the absence of the protein. The following paragraph was added to the discussion:

The autosomal recessive inheritance of USH2A and ADGRV1 suggests a loss of function mechanism. No gain of function variant has been reported in Usher syndrome genes and no case of dominance has been observed, i.e. parents of affected children are healthy. In support of this, most of the patients in this study harbored LOF variants. These included the frame-shifting variants, as they were all predicted to undergo NMD (all were located >50 nucleotides from the last exon-exon junction) and thus expected to result in the absence of protein instead of an aberrant protein [12].

Point 3: Is there any possibly that USH2A and ADGRV1 proteins may physically interacted or USH2A or ADGRV1 may have some protein-protein interacting binding partners in retinal neurons, and the mutant proteins may cause these protein-protein interactions interrupted and cause the loss function of their interacting proteins. If so, please give a brief discussion.

Answer 3: Thenk you for this interesting point. Although a direct interaction has not yet been proven it seems likely. The following paragraph has been added to the discussion:

Considering the proteins' shared location there is a possibility that the proteins physically interact with eachother, and that the loss of function of either protein could result in a dyscunction of both. The possible interaction between USH2A and ADGRV1 has been proposed schematically by Maerker et al [28] however has not yet been proven. It is also possible that the loss of either protein disrupts the interactions between other USH proteins in a similar manner. An existance of a quarternary USH2 complex has been proposed, where the third USH2 protein, whirlin (WHRN), and PDZD7, an USH2 modifier, are required for a complex formation with USH2A and ADGRV1. In this model, WHRN prefers to bind to USH2A, whereas PDZD7 prefers to bind to ADGRV1, while interaction between WHRN and PDZD7 is the bridge between USH2A and ADGRV1 [32]

Point 4: On line 92, the median value was 65 years for USH2A, but could be determined for ADGRV1 due to …, is it possible that “but could be determined …’ should be changed to “but could not be determined …”?

 Answer 4: Thank you, you are correct, the sentence was corrected as suggested.